# The Impact of Apical Periodontitis and Endodontic Treatment on Salivary Inflammatory Biomarkers: A Longitudinal Study

Abdulaziz Bakhsh [1], Noor Al-Abdulla [2], Francesco Mannocci [2], Marwa Allihaibi [2], David Moyes [3], Gordon Proctor [3] and Sadia Ambreen Niazi [2,*]

1 Department of Restorative Dentistry, Division of Endodontics, Faculty of Dental Medicine, Umm Al-Qura University, Makkah 24382, Saudi Arabia

2 Department of Endodontics, Centre of Oral Clinical & Translational Sciences, Faculty of Dentistry, Oral & Craniofacial Sciences, Guy's Dental Hospital, King's College London, London SE1 9RT, UK

3 Centre for Host-Microbiome Interactions, Faculty of Dentistry, Oral & Craniofacial Sciences, Guy's Dental Hospital, King's College London, London SE1 9RT, UK

* Correspondence: sadia.niazi@kcl.ac.uk; Tel.: +44-(0)207-188-7459

**Abstract:** This study aimed to assess the effect of apical periodontitis and its treatment on the profile of salivary inflammatory markers and to investigate its correlation with serum inflammatory markers. Saliva samples were collected from 115 recruited participants. Patients were reviewed after 1 and 2 years following treatment. Saliva samples were analysed using Multiplex microbead immunoassay for identifying the inflammatory biomarkers' profile. Biomarker levels were compared against healthy controls at baseline. Longitudinal comparison of those markers was further analysed for the review appointments and correlated with the size of the periapical radiolucency, treatment outcome and serum inflammatory biomarker levels. The salivary cytokines, matrix metalloproteinases (MMPs) and vascular adhesion molecules were higher at the review appointments. Pre-operative salivary levels of high-sensitivity C-reactive protein (hs-CRP) were significantly higher in the treatment group than in the control group ($p < 0.001$). At 1 year, hs-CRP was decreased than baseline. While, in 2 years, fibroblast growth factor (FGF)-23 was significantly lower compared to baseline levels ($p = 0.005$). Furthermore, the post-operative size of radiolucency was significantly correlated with the levels of several markers. When correlating the salivary levels of biomarkers with the serum levels, a significant correlation was seen in FGF-23 ($p = 0.04$) at baseline; in intercellular adhesion molecule (ICAM)-1 ($p = 0.02$) at 1 year post-treatment; and in TNF-α, ICAM-1 and E-Selectin at 2 years post-treatment ($p = 0.046$; $p = 0.033$; $p = 0.019$, respectively). Therefore, his study suggests that higher salivary cytokines, MMPs and vascular adhesion molecules at the post-treatment reviews are related to periapical bone healing and remodelling, whereas salivary FGF-23 and hs-CRP could be prognostic biomarkers. Correlation of some salivary with serum biomarkers suggests that saliva sampling could be a feasible non-invasive option for the measurement of inflammatory marker levels; however, further longitudinal studies are required.

**Keywords:** apical periodontitis; salivary inflammatory biomarkers; cardiovascular diseases; multiplex luminex immunoassay

## 1. Introduction

Diseases of the oral cavity elicit a local immune response causing the release of inflammatory mediators. Apical periodontitis affecting periapical tissues around the tooth is a chronic inflammatory condition of the oral cavity [1] with a reported global prevalence of almost 52% according to Tibúrcio-Machado, et al. [2]. Bacteria and their byproducts associated with apical periodontitis induce activation of the local immune response, which could contribute to local low-grade chronic inflammation with ensuing localised periapical bone destruction, subsequently affecting systemic health [3].

Oral fluids have previously been collected to determine levels of inflammatory markers associated with various oral and systemic inflammatory conditions [4–7] and can give diagnostic information in oral diseases. Several studies have used saliva as a potential diagnostic tool in autoimmune diseases such as Sjögren's syndrome [8] and cystic fibrosis [9]. Furthermore, salivary levels of C-reactive protein (CRP)—an acute inflammatory marker associated with infections and inflammatory conditions including atherosclerosis—are correlated with plasma levels in patients at risk of cardiovascular complications [10]. Studies have suggested that gingival inflammation may be associated with salivary biomarkers [7], whilst Rathnayake, Akerman, Klinge, Lundegren, Jansson, Tryselius, Sorsa and Gustafsson [4] found that salivary levels of Interleukin (IL)-8, IL-1β and MMP-8 were high in patients with cancer, diabetes, muscle and joint diseases and post-cardiac surgery [4].

Saliva samples can be obtained easily and non-invasively with a minimal risk of cross-contamination, requiring less manipulation than blood samples. In chronic periodontitis, locally upregulated levels of cytokines are correlated with higher salivary levels [11,12]. Several studies have investigated the effect of apical periodontitis and its treatment on serum levels of inflammatory markers associating it with systemic diseases [13–16]. In our previous study, we investigated the effect of apical periodontitis and endodontic treatment on the serum levels of inflammatory markers and showed raised serum inflammatory markers associated with apical periodontitis and their reduction at the 1-year post endodontic treatment follow-up [16]. However, to date, there is a lack of information on the effect of apical periodontitis and its treatment on the levels of salivary inflammatory markers. Therefore, the aim of the present longitudinal study was to identify the effect of apical periodontitis and its treatment on the salivary levels of IL-1β, IL-6 and IL-8; tumour necrosis factor (TNF)-α; pentraxin 3; intercellular adhesion molecule (ICAM-1); vascular adhesion molecule (VCAM-1); E-selectin; high-sensitivity (hs-)CRP; fibroblast growth factor (FGF)-23; and matrix metalloproteases (MMP)-2, 8, and 9. These findings were correlated with the size of the periapical radiolucency, treatment outcome and serum inflammatory biomarker levels.

## 2. Materials and Methods

### 2.1. Ethical Approval

This study was carried out under the ethical approval of the London-Hampstead Research Ethics Committee (IRAS project ID 207795) and by the London-Riverside Research Ethics Committee (REC reference: 20/LO/0024). Participants' consent (written) was obtained according to the Declaration of Helsinki.

### 2.2. Sample Size Calculation

Asymmetric Dimethylarginine (ADMA) levels from Cotti, et al. [17] were used for calculating the sample size. For the control group, a mean value of 0.65 (sd 0.1) was used, and for treatment groups, mean value of 0.74 (sd 0.15) was assumed with 80% power at a 5% level of significance [16]. The results indicated that 19 patients are required per group.

### 2.3. Patient Recruitment

Patients were recruited from an endodontic clinic at Guy's Dental Hospital requiring either root canal retreatment (Re-RCT) or periapical surgery (PS). They were selected based on the inclusion/exclusion criteria explained by Bakhsh et al. 2022 [16] (Table 1). There were three participants groups (Re-RCT, PS and controls). The controls were healthy individuals with no chronic inflammatory condition, no antibiotics taken in the last three months, no surgery in the past six months, no gingivitis or periodontitis and no teeth with apical periodontitis or root canal treatment.

**Table 1.** Inclusion and exclusion criteria [16].

| Inclusion | Exclusion |
|---|---|
| >18 years old | Smokers |
| | Pregnant women |
| | Teeth with periodontal pockets >4 mm/endodontic—periodontal lesion, gingivitis, periodontitis |
| Root canal retreatment cases | Patients with chronic inflammatory condition (Asthma, Inflammatory bowel diseases, Irritable bowel syndrome, Chronic peptic ulcer, Rheumatoid arthritis, Ulcerative colitis, Liver diseases, Crohn's disease, Sinusitis, Active hepatitis, Autoimmune diseases, Tuberculosis, Renal diseases or Cancer). |
| Periapical surgery cases | Patients on medication altering bone metabolism |
| | Unrestorable teeth |
| | Antibiotics in last 3 months |
| | Surgical procedure in last 6 months |

*2.4. Clinical Examination*

During consultation, the patient's chief complaint and detailed history (dental, medical, social) was recorded. Extraoral and intraoral clinical examinations were carried out. Tooth health was investigated using palpation and percussion, along with sensibility testing. A periapical radiograph using a beam-aiming device X-ray unit operating at 65 kV and 7 mAs (Heliodent; Sirona, Benshein, Germany), as well as photostimulable phosphor plates (Digora Optime, Soredex, Tuusula, Finland), was taken at baseline and every review appointment. In addition, Cone Beam Computed Tomography (CBCT) scans (small volume (40 mm$^3$); exposure setting $-90$ kV and 5.0 mAs.) were also taken (3D Accuitomo F170; J Morita Manufacturing, Kyoto, Japan) at baseline and 1-year review. Due to COVID-19 disruption, CBCT scans could not be taken at the 2-year review. For the control group, a detailed history including both medical and dental was recorded along with extraoral and intraoral clinical examinations and panoramic radiograph.

*2.5. Radiographic Outcome Analysis*

Two experienced endodontic specialists reviewed the periapical radiographs and CBCT scans. Treatment outcome scores were recorded after consensus agreement between both examiners using a six-point classification [18].

*2.6. Saliva Sample Collection*

Unstimulated pre-operative saliva samples were collected from treatment groups in a polystyrene 30 mL universal tube (Fisher Scientific, Waltham, MA, USA) for 10 min. Samples were transported in ice and stored at $-80\ °C$ for further analysis.

After saliva collection, the endodontic treatment including root canal retreatment or periapical surgery treatment was performed as detailed in our previous study [16]. Patients were reviewed at 1-year (1 Yr) and 2-year (2 Yr) follow-up after treatment. Medical history was recorded, which included COVID-19 and vaccination status. Changes in the size of periapical radiolucency were investigated using periapical radiographs and CBCT scan (only for 1-year review). Saliva samples were collected at these reviews for salivary biomarker investigation.

Saliva samples collection from the control group at baseline and after 1 year was conducted using the same protocol as described above for treatment patients.

*2.7. Saliva Samples Processing*

After being transported to the lab, saliva samples were centrifuged for 10 min at a speed of $2000\times g$ and a temperature of 4 °C using Eppendorf centrifuge 5810R (Eppendorf,

Hamburg, Germany). The supernatant was stored in a $-80\ °C$ freezer in 0.5 mL aliquots using a 1.5 microcentrifuge tube (Fisher Scientific, Waltham, MA, USA). The quantification of salivary inflammatory markers levels (FGF-23, IL-1β, IL-6, IL-8, hs-CRP, pentraxin 3, TNF-α, MMP-2, MMP-8, MMP-9, E-Selectin, VCAM-1, ICAM-1) was carried out by means of the Magnetic Assay human premixed multi-analyte kit (R&D systems, Bio-techne, Minnesota, MN, USA) using the Bio-Rad Bio-Plex 200 analysers (Bio-rad, Hercules, CA, USA) according to the manufacturer's instructions.

### 2.8. Statistical Analysis

Statistical analyses were done using IBM® SPSS® (Version 15.0). To measure the differences between the same group, the Wilcoxon Signed-Rank test was carried out. However, for multiple comparison, Bonferroni's criteria were used to correct *p*-values. Moreover, differences between two different groups were analysed using the Mann–Whitney test. The level of significance was set to $p < 0.05$, and a 95% confidence interval was generated using a one-sample *t*-test.

The values were converted to a logarithmic scale because of the lack of normality in biomarker levels. The difference between the median of the control and the disease groups was calculated using a two-sample *t*-test. To investigate the linear association with other continuous variables, Pearson's correlation was used; whereas non-linear associations were assessed using Spearman´s correlation. Furthermore, regression models were conducted to estimate beta coefficients involved in those significant correlations. The level of significance was set to $p < 0.05$. In addition, the inter-examiner reliability for PA radiographs was investigated using the linearly weighted Kappa´s index (κ) to assess the concordance.

## 3. Results

In this study, 115 participants were recruited ($n = 50$ controls, $n = 65$ patients with apical periodontitis). From the 65 patients with apical periodontitis (24 males, 41 females; mean age: 43.3 (24–75)), 35 teeth in 35 patients had root canal retreatment, and 30 teeth in 30 patients had periapical surgery. The control group consisted of 50 participants (18 males, 32 females; mean age: 32.4 (19–51)).

### 3.1. Treatment Outcome

Out of 65 patients, 76.9% ($n = 50$) participants were reviewed at 1-year post-treatment. Periapical radiographs assessment showed that 38% ($n = 19$) of the patients completely healed, 54% ($n = 27$%) were healing and 8% ($n = 4$) had failed. At 2-year post-treatment follow-up, 56.9% ($n = 37$) participants were reviewed. Out of these 37 patients, 56.8% ($n = 21$) were completely healed, while 43.2% ($n = 16$) were still healing.

There was substantial agreement between readings of examiners for periapical radiographs at 1-year post-treatment (0.84 (0.69–0.98)) and at 2-year post-treatment (0.72 (0.50–0.94)) follow-ups.

### 3.2. Inflammatory Markers
#### 3.2.1. Pre-Operative Levels

Pre-operative levels of hs-CRP of the diseased group were significantly higher than the control group ($p < 0.001$). The control group had significantly higher levels of TNF-α ($p = 0.029$), IL-6 ($p = 0.005$) and VCAM-1 ($p = 0.005$) (Table 2).

#### 3.2.2. One-Year Post-Treatment Follow-Up

The salivary levels of FGF-23 ($p = 0.0004$), IL-1β ($p = 0.0001$), IL-8 ($p = 0.0005$), TNF-α ($p = 0.003$), ICAM-1 ($p = 0.0021$), IL-6 ($p = 0.002$), E-selectin ($p < 0.0001$), VCAM-1 ($p < 0.0001$), MMP-9 ($p < 0.0001$), MMP-2 ($p < 0.0001$), MMP-8 ($p < 0.0001$) and pentraxin 3 ($p < 0.0001$) were significantly higher at 1-year follow-up than baseline; however, the levels of CRP were reduced but not significantly (Figure 1).

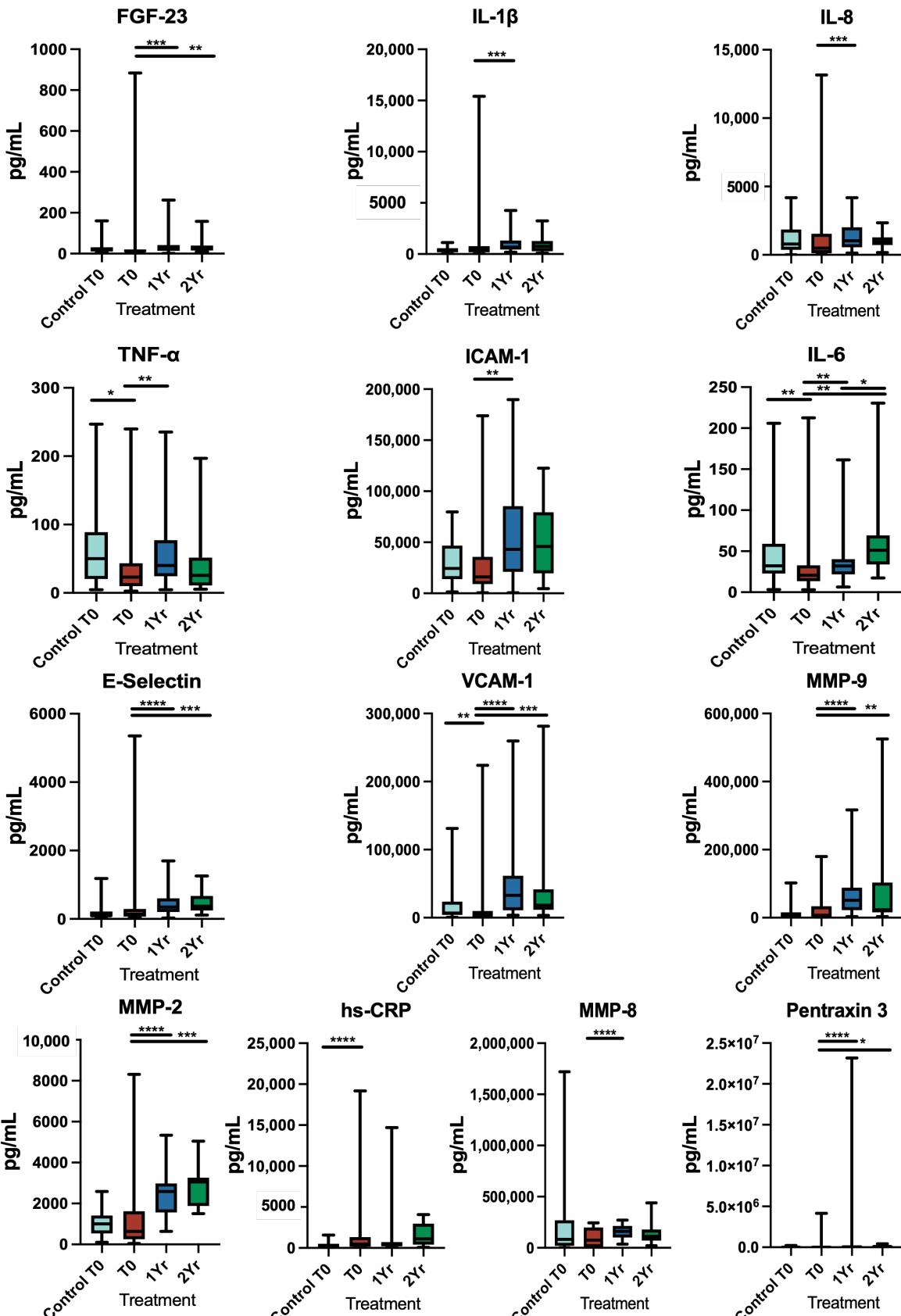

**Figure 1.** Box plot showing salivary levels of inflammatory markers of treatment group at baseline (T0), one year (1 Yr), and two years (2 Yr); differences between medians were compared using Wilcoxon test; * (*p* < 0.05), ** (*p* < 0.01), *** (*p* < 0.001), **** (*p* < 0.0001).

**Table 2.** Salivary inflammatory biomarker level comparison between control and treatment groups at baseline.

| | Control | Treatment | p-Value |
|---|---|---|---|
| | Median (IQR) | Median (IQR) | |
| FGF-23 | 20.44 (10.71–31.26) | 13.23 (8.09–20.17) | 0.089 |
| IL-1β | 354.14 (181.2–577.2) | 398.01 (178.3–772.1) | 0.257 |
| IL-8 | 797.57 (370.2–1860) | 511.09 (117.3–1547) | 0.172 |
| Pentraxin 3 | 4217.6 (2131–8020) | 2093 (1286–29,057) | 0.143 |
| TNF-α | 40.15 (20.28–88.88) | 22.96 (9.79–43.46) | 0.029 * |
| ICAM-1 | 25,883.02 (13,974–46,718) | 15,975.95 (9041–35,671) | 0.144 |
| IL-6 | 31.17 (22.74–59.01) | 20.68 (13.44–32.85) | 0.005 ** |
| MMP-8 | 82,732.46 (18,560–269,023) | 84,431.78 (5458–199,330) | 0.136 |
| E-selectin | 95.22 (65.03–217.3) | 138.86 (68.16–291.4) | 0.182 |
| VCAM-1 | 6412.39 (4205–23,601) | 3059.68 (1557–9886) | 0.005 ** |
| hs-CRP | 192.04 (100.9–502.2) | 372.35 (166.6–1324) | <0.001 *** |
| MMP-2 | 974.52 (537.9–1413) | 638.66 (251.3–1617) | 0.127 |
| MMP-9 | 4446.12 (921.1–15,776) | 8100.99 (2157–33,650) | 0.057 |

*p*-value—difference between median of control and disease group using the Mann–Whitney test; * ($p < 0.1$); ** ($p < 0.01$); *** ($p < 0.001$). IQR: Interquartile range. FGF: Fibroblast Growth Factor; IL: Interleukin, TNF: Tumor Necrosis Factor; ICAM: Intercellular adhesion molecule; MMP: Matrix Metalloprotease; VCAM: Vascular cell adhesion molecule; hs-CRP: high-sensitive C-reactive protein

### 3.2.3. Two-Year Post-Treatment Follow-Up

When the patients were reviewed at two-year post-treatment follow-up, FGF-23 was significantly lower than the pre-operative levels ($p = 0.005$) (Figure 2), whilst the levels of pentraxin 3, IL-6, E-selectin, VCAM-1, MMP-2 and MMP-9 were significantly increased at 2-year post-treatment follow-up ($p = 0.013$; $p = 0.002$; $p = 0.002$; $p = 0.001$; $p = 0.001$; and $p = 0.004$, respectively). Only IL-6 was significantly higher at 2-year ($p = 0.024$) compared to 1-year post-treatment follow-up (Figure 1).

### 3.3. Effect of the Size of Radiolucency on the Levels of Inflammatory Markers

CBCT scans were used to measure the size of radiolucency associated with the teeth requiring treatment. The widest measurement either bucco-lingually or mesio-distally was used, and the size of radiolucency was categorised as <3 mm, 3–5 mm or >5 mm. The pre-operative size of the radiolucency was significantly negatively correlated with the pre-operative levels of FGF-23 ($p = 0.033$) (Figure 2a). An additional millimetre is associated with a decrease in the levels of FGF-23 by 6.5%. However, the 1-year post-treatment lesion size was significantly positively correlated with the levels of FGF-23 ($p = 0.004$), TNF-α ($p = 0.016$) and VCAM-1 ($p = 0.003$) (Figure 2b–d). An increase in the size of the radiolucency by 1 mm correlated with an increase in the levels of FGF-23 by 22.5%, TNF-α by 15.3% and VCAM-1 by 22.5%.

### 3.4. Correlation between the Outcome of Treatment and the Levels of Salivary Inflammatory Markers

Assessing the treatment outcome using PA radiographs showed a significant correlation at 1-year post-treatment follow-up, with the salivary levels of VCAM-1 ($p = 0.01$); the better the outcome, the lower the value of VCAM-1. No significant correlation was evident with CBCT or the 2-year treatment outcome.

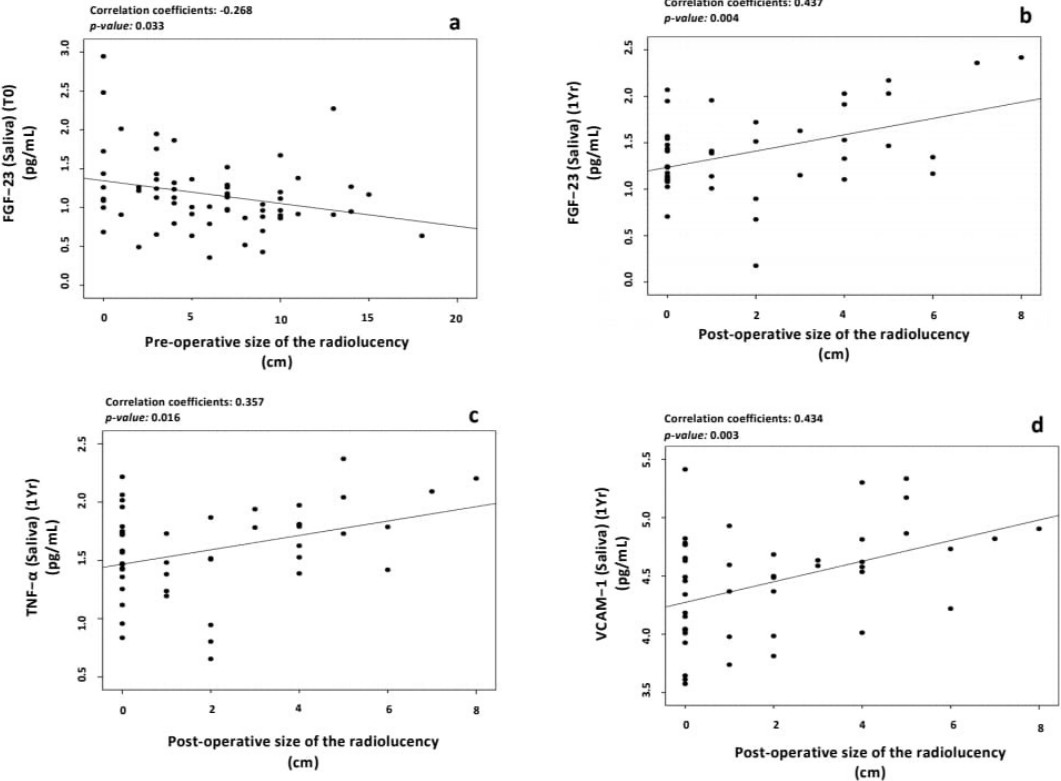

**Figure 2.** Correlation plot showing correlation between pre-operative (T0) and post-operative (1 Yr) salivary levels of inflammatory markers and the pre- and post-operative size of the radiolucency. (**a**) Correlation plot between pre-operative size of radiolucency and the pre-operative salivary levels of FGF-23; (**b**) Correlation plot between post-operative size of radiolucency and the 1−year salivary levels of FGF-23; (**c**) Correlation plot between post-operative size of radiolucency and the 1−year salivary levels of TNF-α; (**d**) Correlation plot between post-operative size of radiolucency and the 1−year salivary levels of VCAM-1.

### 3.5. Correlation between Salivary and Serum Levels of Biomarkers

### 3.5.1. Baseline Levels

Pearson's correlation was used to determine the correlation between the salivary of inflammatory markers and previously reported serum levels [16]. The findings showed that there was a weak direct correlation of the serum levels with of salivary levels of FGF-23 ($p$ = 0.04). In other words, increased levels of saliva by one log unit (pg/mL) would increase the serum levels by 0.23 log unit or 1.69 pg/mL (Figure 3a).

### 3.5.2. One-Year Post-Treatment Follow-Up

At one-year post-treatment, the salivary levels of ICAM-1 were positively correlated with the serum levels of ICAM-1 ($p$ = 0.02). Therefore, an increased level in saliva corresponded to an increased level in serum (Figure 3b).

### 3.5.3. Two-Year Post-Treatment Follow-Up

Salivary levels of TNF-α, ICAM-1 and E-Selectin were positively correlated with the serum levels at two-year post-treatment follow-up ($p$ = 0.046; $p$ = 0.033; $p$ = 0.019, respectively). An extra unit in the salivary levels of TNF-α would increase the serum levels by 1.78 pg/mL, while an increase in ICAM-1 would cause an increase in the serum levels of the same marker by 4.57 pg/mL. Finally, an increase of 1 pg/mL in salivary levels of E-selectin increased the serum levels by 2.14 pg/mL (Figure 3c–e).

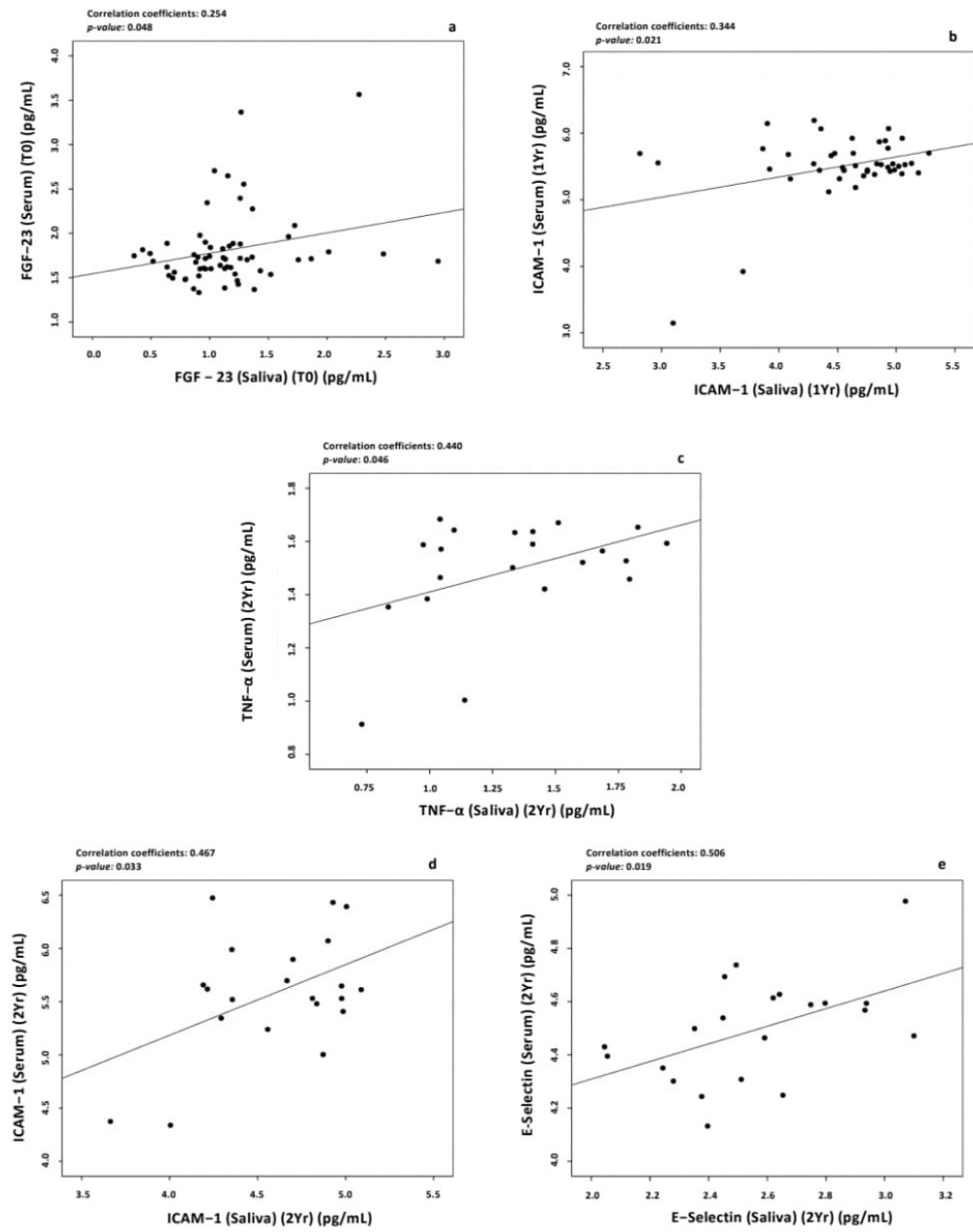

**Figure 3.** Correlation plot between salivary and serum levels of inflammatory markers: (**a**) Correlation between levels of FGF-23 at baseline; (**b**) Correlation between levels of ICAM-1 at 1-year post-treatment follow-up; (**c**) Correlation between levels of TNF-α at 2-year post-treatment follow-up; (**d**) Correlation between levels of ICAM-1 at 2-year post-treatment follow-up; (**e**) Correlation between levels of E-selectin at 2-year post-treatment follow-up.

## 4. Discussion

In our previous study, we investigated the serum inflammatory biomarker levels associated with apical periodontitis and found that successful endodontic treatment reduces the systemic inflammatory burden and thus reduces the risk for CVD development [16]. Salivary sampling can be used to detect diagnostic and prognostic biomarkers for both local and systemic inflammation and offers a non-invasive, low-cost and easy alternative to serum sampling for both patients and researchers [19,20]. The present prospective longitudinal study assessed salivary inflammatory biomarkers' levels in the same cohort of patients as our previous study [16], offering an opportunity to determine changes in local biomarkers associated with apical periodontitis and endodontic treatment and to

investigate any correlation of salivary markers with systemic serum levels. We found that levels of salivary biomarkers were much higher compared to the serum levels, which is probably due to local diffusion of inflammatory cytokines produced in apical periodontitis lesions. Furthermore, we saw that, at the 1-year review appointment, the salivary levels of most of the biomarkers except hs-CRP were still raised compared to the baseline values, whereas in our previous study [16], most of the serum levels did decline at the 1-year review. This may be due to the local influence of healing, potentiating these levels in saliva when the serum levels have already reduced. High levels of these cytokines, MMPs and vascular adhesion molecules in saliva detected at review appointments could be potentially due to their activity in periapical healing by regeneration and bone remodelling [21]. This is consistent with other studies where levels of interleukins and MMPs were raised in saliva and gingival crevicular fluid (GCF) in patients with chronic periodontitis [22,23]. One of the limitations of this study is that GCF was not investigated. This can provide more accurate information about the local inflammation and answer why some markers in saliva were still higher after endodontic treatment when serum markers had already reduced. Investigating correlations of GCF biomarkers with serum and salivary biomarkers levels can provide information about the reliability of salivary biomarkers for endodontic treatment.

The one-year recall rate was 76.9%, which was reduced at 2-year post-treatment review due to COVID-19 restrictions. It has been found that, in England, almost 19 million dental appointments have been missed in the NHS due to the restrictions [24]. However, the recall rate of our study was comparable to similar outcome studies [25,26]. The overall treatment outcome when considering the healed and healing cases using periapical radiographs was 92% at 1-year review, which reached 100% at 2-year post-treatment follow-up. This emphasises that care should be taken when assessing the outcome, as the European Society of Endodontics recommends follow-up of patients with root-treated teeth for up to four years post-treatment [27].

The pre-operative levels of hs-CRP in the apical periodontitis group were significantly higher at baseline when compared to the control group. At 1-year post-treatment follow-up, hs-CRP salivary levels were reduced compared to pre-operative levels. CRP is an acute phase protein produced by the liver and induces the production of pro-inflammatory cytokines [28,29]. CRP levels could potentially be used as a diagnostic marker for the prediction of CVDs and myocardial infarction [10,30]. However, Pay and Shaw [31] found that the salivary levels of CRP are not as reliable and do not strongly correlate with serum levels. This may be due to the presence of local oral inflammation, such as gingivitis, contributing to its increased local production. In our study, saliva showed higher levels of hs-CRP compared to serum but in both saliva and serum; the hs-CRP levels decreased at 1-year post-treatment follow-up compared to their respective baseline.

Levels of hs-CRP have been shown to be an indicator of systemic inflammation when tested in the gingival crevicular fluid (GCF) rather than the result of local production by gingival cells [32]. As hs-CRP is only produced by the liver, it has been found that it enters saliva through passive diffusion or filtration from other oral fluids, such as GCF, resulting in raised levels in saliva [33]. Studies have shown that salivary levels of CRP are strongly correlated with serum levels in adults. Specifically, elevated salivary levels of CRP were correlated with serum directly after acute myocardial infarction. Therefore, salivary hs-CRP may be a viable option for the detection of acute myocardial infarction in adults [34]. However, this correlation would depend upon the level of oral inflammation.

Several studies have investigated the correlation of salivary and serum biomarkers. For example, Byrne, et al. [35] found a medium correlation between salivary and serum samples for the detection of CRP, TNF-$\alpha$ and IL-10 [35]. Furthermore, Cullen, et al. [36] found a correlation in the levels of IL-6 in both serum and saliva. In our study, a positive correlation was found between the salivary and serum levels of FGF-23, ICAM-1, E-Selectin and TNF-$\alpha$, indicating the possible use of saliva as an inflammatory indicator as a substitute to serum.

FGF-23 is a hormone produced by osteocytes and osteoblasts and, therefore, any bone alteration caused by inflammation could indirectly affect the production of FGF-23 [37,38]. Increased levels of FGF-23 were found to impact the kidney and the heart by increasing the sodium absorption, renin-angiotensin activation and affecting the heart and blood vessels, which subsequently may lead to hypertension, subclinical atherosclerosis and cardiovascular events [39,40]. Furthermore, FGF-23 is produced in lower concentrations in salivary glands, stomach, skeletal muscles, brain, liver and heart [41]. Salivary levels of FGF-23 were reduced after 2 years when compared to 1-year follow-up, and a positive correlation was seen between pre-operative salivary levels of FGF-23 and the post-operative size of the radiolucency. It has been found that excess levels of FGF-23 can negatively impact bone mineralization [42]. When analysed in serum, FGF-23 was the only marker that was reduced in all review appointments, indicating its potential use as a treatment outcome predictor [16].

Finally, soluble adhesion molecules including ICAM-1 and VCAM-1 were found to have an impact in the progression of atherosclerosis and CVDs [43]. ICAM-1 is a glycoprotein interfering with the adhesion-dependent cell-to-cell interaction. Both ICAM-1 and VCAM are activated by cytokines including TNF-$\alpha$, reactive oxygen species and high glucose concentration [44]. In our study, the levels of VCAM-1 were positively correlated with the pre-operative size of radiolucency. In addition, at 1-year post-treatment follow-up, it was positively correlated with the treatment outcome (i.e., the better the outcome, the lower the levels of VCAM-1). When correlating the serum and salivary inflammatory markers' levels, ICAM-1 levels of saliva were positively correlation to serum levels. This was in line with the study done by Yılmaz Şaştım, et al. [45] in patients with periodontitis, indicating that the presence of local inflammation, such as apical periodontitis, may have systemic effect and increase systemic burden [45].

## 5. Conclusions

To date, there is lack of evidence on the impact of apical periodontitis and its treatment on the levels of salivary inflammatory markers and it remains unknown if these local levels are associated with their serum systemic levels. Overall, our findings suggest that some biomarkers can be measured in saliva, especially CRP and FGF-23. The higher salivary cytokines, MMPs and vascular adhesion molecules at the post-treatment reviews are related to periapical bone healing and remodelling, whereas salivary FGF-23 and hs-CRP could be prognostic biomarkers. Correlation of some salivary with serum biomarkers suggests that saliva sampling could be a feasible non-invasive option for the measurement of inflammatory markers levels. However, further studies with larger samples investigating gingival crevicular fluid levels of these biomarkers and correlating these with saliva and serum samples at different time points might give a more accurate indication of the location inflammation and confirm the reliability of using these examined biomarkers, as well as ascertain if saliva may be used as a medium to measure systemic inflammation compared to blood.

**Author Contributions:** Conceptualization, A.B. and S.A.N.; methodology, A.B., N.A.-A. and S.A.N.; validation, A.B., S.A.N., N.A.-A., M.A. and D.M.; formal analysis, A.B., N.A.-A. and M.A.; investigation, A.B. and S.A.N.; resources, A.B., S.A.N., G.P. and D.M.; data curation, A.B. and S.A.N.; writing—original draft preparation, A.B. and S.A.N.; writing—review and editing, A.B., N.A.-A., M.A., S.A.N., D.M., F.M. and G.P.; visualization, A.B., S.A.N., G.P. and D.M.; supervision, S.A.N., G.P., D.M. and F.M.; project administration, A.B. and S.A.N.; funding acquisition, S.A.N. All authors have read and agreed to the published version of the manuscript.

**Funding:** The study is funded by the British Endodontic Society (Grant for Research Work). Sadia Ambreen Niazi is the principal grant holder of this grant.

**Institutional Review Board Statement:** The study was conducted in accordance with the Declaration of Helsinki and approved by the London-Hampstead Research Ethics Committee (IRAS project ID 207795) and by the London-Riverside Research Ethics Committee (REC reference: 20/LO/0024).

**Informed Consent Statement:** Informed consent was obtained from all subjects involved in the study.

**Data Availability Statement:** The data presented in this study are available on request from the corresponding author.

**Acknowledgments:** We would like to thank Juan Luis Gómez Martínez for his help in statistical analysis of the data.

**Conflicts of Interest:** The authors declare no conflict of interest.

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
