# Peer review of "The Impact of Apical Periodontitis and Endodontic Treatment on Salivary Inflammatory Biomarkers: A Longitudinal Study"

_applsci, doi:10.3390/app13063952_

Round 1
Reviewer 1 Report
1. In table 2, it is better to give the interquartile range, rather than the minimum and maximum values, this will be more informative. Does Figure 1 duplicate the data given in Table 2?
2. I would recommend the authors to remove Figure 1, as it repeats the data of Table 2, and add Control in Figure 2. This will make it possible to compare the indicators in dynamics with the control values.
3. The authors give a fairly large number of inflammatory markers that can potentially be determined in saliva. Which one is most suitable for monitoring the inflammatory process after treatment?
Author Response
Dear Reviewer1,
Thank you very much for your review and comments. Please find below our reply.
- In table 2, it is better to give the interquartile range, rather than the minimum and maximum values, this will be more informative. Does Figure 1 duplicate the data given in Table 2?
Interquartile range added, Figure 1 (Removed)
- I would recommend the authors to remove Figure 1, as it repeats the data of Table 2, and add Control in Figure 2. This will make it possible to compare the indicators in dynamics with the control values.
Figure 1 removed, and the Control data was added to figure 1 (figure 2 previously)
- The authors give a fairly large number of inflammatory markers that can potentially be determined in saliva. Which one is most suitable for monitoring the inflammatory process after treatment?
After the treatment, MMPs are suitable for monitoring as they are locally secreted in the saliva and they are involved in the bone healing process. The results of our study also showed that salivary hs-CRP and FGF-23 reduced at 1 year and 2 year reviews respectively, so these might also be markers related to healing after treatment, however further studies are required as we explained in our conclusion.
Reviewer 2 Report
Review for Manuscript ID: applsci-2275428 entitled " The impact of Apical Periodontitis and Endodontic Treatment on Salivary Inflammatory Biomarkers: A Longitudinal Study”
The manuscript is well designed, but there are points need to be addressed as follows:
1. Abstract: ICAM abbreviation need to be defined.
2. Abstract: the sentence “Therefore, the result of this study suggests that saliva sampling could be a feasible 28 non-invasive option for the measurement of inflammatory markers levels” as conclusion is not in line and do not answer the study title and aim. The study aim and title looking to the impact of apical periodontitis and treatment on salivary biomarkers.
3. Reference no. 2. Need to be corrected
4. I think this paper better highlight the importance of oral fluid as diagnostic tool than reference 8-10 (Gul, S.S.; Abdulkareem, A.A.; Sha, A.M.; Rawlinson, A. Diagnostic Accuracy of Oral Fluids Biomarker Profile to Determine the Current and Future Status of Periodontal and Peri-Implant Diseases. Diagnostics 2020, 10, 838. https://doi.org/10.3390/diagnostics10100838).
5. I think the data of root canal retreatment (Re-RCT) or periapical surgery (PS) have to be presented differently as these treatment choices are completely different and thus their outcome (clinical, radiograph and salivary impacts).
6. The primary outcome measure, healing is not clear, elaboration is needed.
7. it is not clear that patients in test group have only one tooth to receive endodontic treatmetnt.
8. Table 2, instead of min-max with median, it is necessary to report the interquartile range (25% - 75%). Also, unit of each biomarker need to be added beside the name if the biomarker
9. I think figure 1, is repetition of table 2 and has to be removed.
10. Figure 2, has to be presented as table to allow the reader to better see the data.
11. Radiographs (CBCT) before and after treatment (1 year and 2 year) have to be included in the result section.
12. The author report serum level of biomarkers (figure 4) which serum collection and analysis is not mentioned in method section.
13. in method section: selection for the tooth for endodontic treatment is not mentioned. This has to be very clear as inclusion and inclusion criteria.
14. Conclusions: need to be rewritten as it does not answer the study’s question. Things about previous study have to be omitted. Clear conclusion about the impact of endodontic treatment on the examined biomarkers need to be stated.
15. To me, it does look that these biomarkers are not reliable for endodontic treatment as they are even higher after treatment. This need to be clear in the conclusion and properly discussed in the discussion section.
16. why the author did not use GCF as it does more reflect the local inflammation and might answer why the examined biomarkers are not that reliable.
17. it is not clear in the method section whether patients with periodontal diseases such as gingivitis and periodontitis included and how standardized. Periodontal diseases do impact on salivary biomarkers and therefore might be behind the high level of the biomarkers after treatment. This need to be discussed.
BW,
Author Response
Dear Reviewer2,
Thank you very much for your review and comments. Please find below our reply.
The manuscript is well designed, but there are points need to be addressed as follows:
- Abstract: ICAM abbreviation need to be defined.
Added
- Abstract: the sentence “Therefore, the result of this study suggests that saliva sampling could be a feasible 28 non-invasive option for the measurement of inflammatory markers levels” as conclusion is not in line and do not answer the study title and aim. The study aim and title looking to the impact of apical periodontitis and treatment on salivary biomarkers.
Thank you for pointing this out. The aims and conclusion has been restated and are now aims and conclusion in abstract are aligned.
- Reference no. 2. Need to be corrected
Corrected
- I think this paper better highlight the importance of oral fluid as diagnostic tool than reference 8-10 (Gul, S.S.; Abdulkareem, A.A.; Sha, A.M.; Rawlinson, A. Diagnostic Accuracy of Oral Fluids Biomarker Profile to Determine the Current and Future Status of Periodontal and Peri-Implant Diseases. Diagnostics 2020, 10, 838. https://doi.org/10.3390/diagnostics10100838).
Reference changed.
- I think the data of root canal retreatment (Re-RCT) or periapical surgery (PS) have to be presented differently as these treatment choices are completely different and thus their outcome (clinical, radiograph and salivary impacts).
We agree that they are very different treatment modalities, but we are looking at their effect locally. Surprisingly, our previous study showed that the systematic inflammatory effect on the levels of serum inflammatory markers for both these treatments were similar (Bakhsh et al. 2022). Therefore, in this saliva study we have presented them together.
Bakhsh, A.; Moyes, D.; Proctor, G.; Mannocci, F.; Niazi, S.A. The Impact of Apical Periodontitis, Non-surgical Root Canal Retreatment and Periapical Surgery on Serum Inflammatory Biomarkers. International Endodontic Journal 2022, 10.1111/iej.13786, doi:10.1111/iej.13786.
- The primary outcome measure, healing is not clear, elaboration is needed.
Sentence changed to “Changes in the size of periapical radiolucency”.
- it is not clear that patients in test group have only one tooth to receive endodontic treatment.
Elaborated in the results section.
- Table 2, instead of min-max with median, it is necessary to report the interquartile range (25% - 75%). Also, unit of each biomarker need to be added beside the name if the biomarker
The IQR has been added to table 2
- I think figure 1, is repetition of table 2 and has to be removed.
Figure 1 has been removed
- Figure 2, has to be presented as table to allow the reader to better see the data.
The control data has been added to figure 1 (previously figure 2) as requested by reviewer 1
- Radiographs (CBCT) before and after treatment (1 year and 2 year) have to be included in the result section.
Due to covid disruptions, CBCT scan were not possible at 2-year review for all patient, therefore CBCT data was insufficient at 2 -year and therefore not included. We have included the radiographic results (outcome) using periapical radiographs at all time points in results section.
- The author report serum level of biomarkers (figure 4) which serum collection and analysis is not mentioned in method section.
The serum data are presented in our previous study (Bakhsh, A.; Moyes, D.; Proctor, G.; Mannocci, F.; Niazi, S.A. The Impact of Apical Periodontitis, Non-surgical Root Canal Retreatment and Periapical Surgery on Serum Inflammatory Biomarkers. International Endodontic Journal 2022, 10.1111/iej.13786, doi:10.1111/iej.13786.)
- in method section: selection for the tooth for endodontic treatment is not mentioned. This has to be very clear as inclusion and inclusion criteria.
No selection criteria was determined, teeth requiring root canal retreatment were included
- Conclusions: need to be rewritten as it does not answer the study’s question. Things about previous study have to be omitted. Clear conclusion about the impact of endodontic treatment on the examined biomarkers need to be stated.
Conclusion has been modified.
- To me, it does look that these biomarkers are not reliable for endodontic treatment as they are even higher after treatment. This need to be clear in the conclusion and properly discussed in the discussion section.
Thank you for your suggestion. This has been added in the discussion section.
- why the author did not use GCF as it does more reflect the local inflammation and might answer why the examined biomarkers are not that reliable.
Thank you for your suggested. We have modified our conclusion accordingly.
- it is not clear in the method section whether patients with periodontal diseases such as gingivitis and periodontitis included and how standardized. Periodontal diseases do impact on salivary biomarkers and therefore might be behind the high level of the biomarkers after treatment. This need to be discussed.
In this longitudinal study our exclusion criteria were patients with gingivitis and periodontitis. The periodontal assessment was carried out at every patient appointment. Patient with periodontal disease were not included in this study.
Round 2
Reviewer 1 Report
The authors took into account the comments of the reviewers and substantially revised the manuscript. I think that in its present form the article can be recommended for publication.
Author Response
Thank you very much for reviewing the comments.
Reviewer 2 Report
NIL
Author Response

(The authors gave the same response as above.)
